# Exploring Hyperdimensional Computing for Anomaly-Based Intrusion Detection Systems

## Abstract

Traditional Intrusion Detection Systems (IDS) are often inefficient at protecting Internet of Things (IoT) devices due to the computational constraints of these devices and the evolving nature of cyberattacks. Although classical Machine Learning (ML) algorithms such as Random Forest and Decision Tree achieve high accuracy, their computational cost and memory consumption make them impractical for large-scale deployment in resource-constrained environments. To tackle this issue, this research investigates the use of Hyperdimensional Computing (HDC), a bio-inspired and energy-efficient alternative, as the core of an IDS for IoT. We propose a comprehensive benchmarking analysis comparing six state-of-the-art HDC models (BinHD, NeuralHD, OnlineHD, AdaptHD, DistHD, CompHD) against six traditional ML algorithms on modern cybersecurity datasets (NSL-KDD, IoT-Flock, UNSW-NB15, BCCC-CIC-IDS2017, CIC-IDS-2017, and BotNetIoT-L01). Our results demonstrate that the most advanced techniques in HDC models are able to improve accuracy while maintaining low memory consumption, especially NeuralHD and AdaptHD, which obtained the best accuracy results among the HDC models. NeuralHD consumed 467x less memory with a 4.72% accuracy difference compared to Random Forest on the UNSW dataset. We conclude that HDC emerges as a viable and necessary paradigm for IoT IDS, providing an ideal trade-off between performance, computational cost, and memory consumption.

## 1 Introduction

There is an abundance of data worldwide. The widespread availability of sensitive information across global digital infrastructures presents a significant challenge in cybersecurity, privacy preservation, and data governance. Confidential information is present in organizations, healthcare, and government data, among other sources (Shinde and Ansurkar, 2023). Given this situation, intrusion detection systems (IDS) are security mechanisms designed to monitor network traffic, detecting malicious activities that attempt to subvert security policies. Therefore, in these circumstances, traditional ML algorithms are used to combat threats and increase security in breach scenarios in IoT device network, such as the work presented in (Hussain et al., 2021) with a specific focus on the healthcare environment, which proposed a labeled dataset containing normal and attack traffic, enabling ML-based attack detection in a testbed setup.

HyperDimensional Computing (HDC) is an emerging technique that enhances and joins power with IDS's capabilities by leveraging high-dimensional hypervectors to analyze data similarity. Beyond that, HDC augments IDS through highperdimensional hypervectors, combining noise-resistant redundancy with uniform information distribution. This approach achieves energy-efficient, adaptive learning and classification (Ge and Parhi, 2020). HyperDetect (Wang et al., 2024) is a real-time innovative HDC solution for IoT intrusion detection inspired by neural mechanisms, which reports the superiority of HDC over conventional SVM and deep learning (DL) in terms of in real-time operation through computational efficiency, noise resilience in unstable environments, and minimal hardware footprint for cutting-edge execution. However, HyperDetect is the only model, to our knowledge, focused on the IDS problem that has been evaluated with only two ML models and two HDC models. There are significant opportunities to investigate and develop HDC models to improve IDS techniques.

This paper aims to investigate how HDC models handle IDS in terms of accuracy, time, and memory consumption using conventional ML libraries such as PyTorch and Scikit-learn. Overall, we evaluated six traditional ML algorithms and six HDC models, comparing their performance on six intrusion detection datasets.

## 2 RELATED WORK

The research community has proposed a wide range of security mechanisms to protect both IoT and conventional networks against malicious intrusions. Over the past years, many Network Intrusion Detection Systems (NIDS) approaches have been developed that take advantage of the strong capabilities of ML and DL models. Fundamental concepts about IDS are discussed in Appendix A.1. In this context, due to the resource constraints of IoT nodes and protocol-specific vulnerabilities (e.g., CoAP, MQTT), conventional IDS solutions are inadequate for IoT environments. Based on this premise, to address these gaps, (Hussain et al., 2021) proposed an open source framework tool, the IoT-Flock, to generate labeled IoT healthcare datasets composed with normal and attack network traffic. The proposed work enables ML-based attacks detection and context-aware security solutions for critical IoT healthcare systems. Following the direction set by established intrusion detection methods, (Zhour et al., 2023) proposes a voting-based ensemble model that integrates Random Forest, Decision Tree, and MLP classifiers to enhance detection performance and efficiency. Their approach emphasizes rapid preprocessing, reduced training time, and high accuracy, achieving 99.7% on NSL-KDD, 77.99% on UNSW-NB15, and 84.89% on CIC-IDS-2017 datasets in multiclass classification tasks.

FlowHacker (Sacramento et al., 2018) introduces an unsupervised machine learning approach to detect novel network attacks by analyzing flow-level traffic data. Unlike signature-based methods, it leverages the imbalance between normal and malicious traffic, clustering flows to isolate anomalies (smaller clusters) for further analysis. The system combines unsupervised clustering with threat intelligence to classify suspicious traffic and identify malicious hosts, eliminating the need for labeled training data.

Regarding deep learning-based intrusion detection, a two-stage hybrid model combining Long Short-Term Memory (LSTM) networks and Autoencoders (AE), referred to as LSTM-AE (Lachekhab et al., 2024), is designed to efficiently handle large volumes of complex raw network data while maintaining high detection accuracy. The model emphasizes dimensionality reduction and feature retention, addressing issues such as overfitting and class imbalance. Evaluated on the CIC-IDS-2017 and CSE-CIC-IDS2018 datasets, the LSTM-AE demonstrated superior performance compared to conventional models, confirming its effectiveness in modern intrusion detection scenarios. A hybrid CNN-LSTM model (Pear and Kibria, 2024) for the detection of network intrusion is proposed by integrating convolutional layers for effective feature extraction with LSTM layers to capture temporal dependencies in network traffic data. The model is tailored for both binary and multiclass classification tasks, utilizing binary and categorical cross-entropy loss functions, respectively. Experimental results on UNSW-NB15 dataset demonstrate strong performance, achieving 97.19% accuracy in binary classification and 87.70% in multiclass scenarios. In particular, the model exhibits robustness in nine attack categories, significantly improving detection accuracy and reducing false positives, highlighting its potential to improve network security.

Recent work (Wang et al., 2024) proposes HyperDetect, a real-time HDC solution for IoT intrusion detection. The groundbreaking work is introduced as the first network intrusion detection model utilizing the parallel processing capabilities of hyperdimensional computing (HDC), inspired by neural mechanisms. The framework incorporates an innovative momentum-aware updating approach for model optimization. The optimized HyperDetect model achieves robust intrusion detection with fewer training cycles and reduced dimensionality, significantly speeding up training and inference by removing computational overhead. CyberHD (Wang et al., 2023a) is an innovative HDC-based framework that improves efficiency by identifying and regenerating insignificant dimensions, enabling effective threat detection with reduced dimensionality. Additionally, its holographic data representation enhances robustness against hardware errors. The authors further optimize this approach using matrix operations, making it suitable for real-time attack detection on resource-constrained devices. Appendix A.2 introduces the fundamental concepts regarding HDC model.

## 3 METHODOLOGY

The binary classification employed in this research differentiates two classes: Attack and Non-Attack. It encompasses data preprocessing, model training, and performance evaluation to effectively identify anomalies. Moreover, to boost the differences between the utilization of HDC and ML model's classification accuracy and pertinent metrics, advanced techniques are applied to capture complex patterns in the data, strengthening their differences. Therefore, the core goal of this study is to compare two types of models and their strengths and weaknesses, in addition to ensuring the most reliable detection of all potential threats, thereby protecting the network from vulnerabilities and reinforcing its security.

### 3.1 DATASET COLLECTION

As previously discussed, numerous datasets exist in the field of network intrusion detection, which can be broadly classified into realistic datasets (collected from real-world systems) and simulated datasets (generated using simulation tools). Consequently, this domain includes attributes with statistical and temporal distributions derived from packet metrics and network data, such as flow duration, source and destination IP addresses, communication protocols, captured in different large-scale network configurations, as well as several other characteristics relevant to network connectivity. Therefore, holistically, with regard to class distribution, the data instances cover a wide spectrum of attack types, but normal traffic prevails in most cases. For this reason, given that the main objective is to detect anomalies using a classifier, the labels of all datasets were stratified into normal and anomalous. Among notable examples, this research utilizes:

1. **NSL-KDD**: The dataset was created to overcome the shortcomings of the KDD Cup 99 dataset. It maintains the same four attack categories but has a more refined structure with distinct training and testing files. The training set has 126,620 instances with 21 attack types, and the test set contains 22,850 instances (Tavallaee et al., 2009).

2. **IoT-Flock**: The dataset by (Hussain et al., 2021) was produced using IoT-Flock, an open-source tool that generates IoT sensor traffic supporting MQTT and CoAP protocols. It simulates a hospital ICU with two beds, each connected to multiple monitoring devices. The tool captures both normal and malicious network behaviors in PCAP (*Packet Capture*) format, later converted into labeled CSV files. The dataset includes features from network, transport, application, and payload layers, such as inter-arrival times, TCP flags, and MQTT control fields. Using logistic regression, the authors identified the ten most significant attributes for machine learning classification.

3. **UNSW-NB15**: The dataset, created by the Australian Cyber Security Centre (ACCS), combines normal and malicious network traffic. It includes nine attack types—such as fuzzers, DoS, exploits, and worms—and is split into training and test sets, both containing a mix of attack and benign samples. The full dataset has over 2.5 million records, with 175,341 in the training subset and 82,332 in the test subset (Moustafa and Slay, 2015).

4. **CIC-IDS-2017**: The dataset, created by the Canadian Institute for Cyber Security, offers realistic network traffic with both normal behavior and modern attacks. Using the CICFlowMeter tool, it captures detailed flow characteristics such as timestamps, IP addresses, protocols, and attack signatures. The dataset covers eight major attack types, including brute force, DoS, DDoS, Heartbleed, web attacks, infiltration, and botnets (Sharafaldin et al., 2018).

5. **BCCC-CIC-IDS2017**: The dataset is a flow-based intrusion detection resource derived from processing the CIC-IDS-2017 dataset using the NTLFlowLyzer framework. This open-source Python tool extracts network-layer features from TCP traffic to support anomaly profiling. Unlike the original raw PCAP files, this refined dataset provides structured CSV files with well-defined flow features from network and transport layers, facilitating easier application of machine learning, behavioral analysis, and anomaly detection techniques for researchers (Shafi et al., 2025).

6. **BotNet-IoT-L01**: The dataset was developed by UNSW Canberra Cyber Range Lab using a realistic network environment that blends normal and botnet traffic. Available in

both PCAP and CSV formats, the dataset is organized by attack category and subcategory—including DDoS, DoS, scanning, keylogging, and data exfiltration—to support accurate labeling. DDoS and DoS attacks are further classified by protocol, enhancing its utility for intrusion detection research (Koroniotis et al., 2019).

## 3.2 DATASET PRE-PROCESSING

At this stage of the research, the experiments were conducted in the Google Colab environment (Bisong, 2019), a cloud computing platform suitable for running Python code and data analysis libraries. Initially, as the datasets were distributed in individual files, a preliminary validation was performed to check columns and instance counts. After concatenation, it was necessary to understand the domain of each dataset through a technical and statistical summary, highlighting the distribution of attributes relevant to classification.

Next, in the cleaning stage, a thorough analysis was conducted to identify inconsistencies, treat missing data, eliminate duplicates and, empty or noisy records, and verify data types (categorical and numerical). This phase was essential to ensure that the algorithms were not affected by biases.

The next step focused on analyzing the target attribute, which, because it is an anomaly-based detection system, was defined in two classes: Normal and Anomalous. The ratio between these classes was verified, noting that, although some datasets were imbalanced, it was not necessary to apply oversampling/undersampling techniques, as the impact did not compromise the main objective of anomaly detection.

Finally, due to the high volume of instances in some datasets, memory and processing (CPU/GPU) usage was optimized by adjusting data types, ensuring efficiency without losing consistency. This entire process aimed to ensure data quality before applying the classification models.

## 3.3 CLASSIFICATION MODELS

Currently, HDC remains primarily a research-focused area, with few real-world applications. However, (Hassan et al., 2022) asserts that the growing volume of research exploring new uses for HDC indicates promising future potential. In this vein, in an effort to strengthen intrusion detection techniques in several environments, this study examines and compares six prominent HDC models, focusing on their core features and operational principles, each with unique strengths, namely, BinHD (Imani et al., 2019a), NeuralHD (Zou et al., 2021), OnlineHD (Hernández-Cano et al., 2021), AdaptHD (Imani et al., 2019b), DistHD (Wang et al., 2023b), and CompHD (Morris et al., 2019). Each HDC model is described in Appendix A.3

The selection and configuration of the ML Algorithms reflect a balanced approach to evaluating diverse algorithmic paradigms, including probabilistic models such as Gaussian Naive Bayes, linear classifiers like Logistic Regression, ensemble methods including Random Forest and AdaBoost, instance-based learners such as K-Nearest Neighbors, and Decision Trees. Each model was configured with carefully chosen hyperparameters designed to mitigate overfitting while maintaining competitive performance.

## 3.4 DATASET PROCESSING

With the dataset prepared, as summarized in Table 1, scenarios were planned for the application of HDC and ML algorithms in binary classification. As highlighted by (Chen et al., 2021), feature selection via Pearson's coefficient ($r \geq 0.7$) significantly reduces the dimensionality of data in intrusion detection systems, optimizing both computational efficiency and classification accuracy. This step is crucial, as intrusion detection datasets often contain redundant or irrelevant attributes.

To eliminate multicollinearity, the correlation matrix was calculated and a Boolean mask (Pearson $r \geq 0.7$) was applied, removing highly correlated column pairs. In addition, Mutual Information (MI) was used to capture nonlinear relationships between variables, following the proposals of (Kraskov et al., 2004) and (Zhao et al., 2018). The data were then standardized with Standard Scaler (Scikit-Learn) to uniform scales, except in the IoT-Flock dataset, which used logistic regression to select the 10 most relevant attributes, according to (Hussain et al., 2021).

Within the phase of data division and evaluation, the processed sets were divided into training (70%) and testing (30%) via Train-Test-Split (Scikit-Learn), a strategy that minimizes overfitting by validating the model with data not seen during training. To ensure robustness, each algorithm (HDC and ML) was run in 30 independent iterations, recording metrics such as accuracy, precision, recall, F1-score, AUC-ROC, as well as processing times and energy consumption (CPU, GPU, RAM). The estimated $CO_2$ emissions and model sizes were also measured, allowing for a comparative analysis of computational cost and energy efficiency. The HDC models were tested with different epochs to assess their impact on performance. This systematic approach aims to ensure the generalization of results and the effectiveness of the models in real scenarios.

## 4 Experiments and Results

This section details the experimental setup, including the datasets and evaluation metrics, and presents a comprehensive and extensive analysis of the results from benchmarking traditional machine learning models against hyperdimensional computing models.

### 4.1 Experimental Setup

The experiments were conducted on a desktop server equipped with Intel® Core™ i7-1070 (8C/16T) @ 2.9 GHz (x86_64) processor, 16 GB DDR4 RAM, NVIDIA GeForce RTX 3060 (Driver 570.133.07, CUDA 12.1), running Ubuntu Linux. All experiments were implemented in Python, and the complete source code is publicly available for reproducibility.

The ML models were implemented using the Scikit-learn library, with the hyperparameters set to the default settings for GaussianNB, AdaBoost, and Logistic Regression. Random Forest was configured with a maximum depth of ten levels, Decision Tree was configured using the Gini index as split classifications, a maximum depth of ten levels, and a requirement of at least five samples per leaf, and KNN was configured with five neighbors.

The HDC models were implemented using the TorchHD (Heddes et al., 2023) library and configured to use 1000-dimensional hypervectors. NeuralHD, OnlineHD, and DistHD use identical retraining mechanisms (regeneration frequency = 20, regeneration rate = 0.04), while AdaptHD and CompHD adopt quantization strategies (n_levels = 100) to handle continuous data. Learning rates vary (0.035–0.37) to reflect different convergence behaviors, and most models utilize GPU acceleration for greater efficiency. Overall, the configuration balances uniformity for fair evaluation with specialized settings that exploit each model's unique strengths.

### 4.2 Comparative Analysis Among ML and HDC Models

To compare the results of the ML and HDC models, the three models with the best accuracy results for the ML approach were chosen as presented in Appendix A.4.1, and the three best models based on HDC were chosen as presented in Appendix A.4.2. Thereby, in relation to accuracy perspective, clearly presented in Figure 1, traditional ML models such as Decision Tree, Random Forest, and KNN consistently achieve very high performance, particularly on datasets like NSL-KDD, BCCC-CIC-IDS-2017, CIC-IDS-2017, and BotNetIoT, where they dominate the comparison. However, the NeuralHD and the AdaptHD models demonstrate competitive performance, reducing the gap significantly in IoT-Flock and UNSW datasets, where they achieve accuracy values close to, and in some cases surpassing, their ML counterparts. OnlineHD follows with slightly lower accuracy levels but still demonstrates robustness across multiple datasets.

In the context of models with lower $CO_2$ emissions as depicted in Figure 2. Regarding the NSL-KDD dataset, the Decision Tree algorithm stands out as the most efficient traditional model, with extremely low emissions, close to $10^{-5}$ kg. In contrast, Random Forest and KNN present much higher values, reaching approximately $10^{-3}$ kg, which reflects the high cost, in the case of KNN, of performing distance calculations in high-dimensional spaces. The models based on HDC appear in the intermediate range, with values close to those of Random Forest, with AdaptHD standing out as the most economical, followed by NeuralHD, while OnlineHD shows the highest consumption within this dataset.

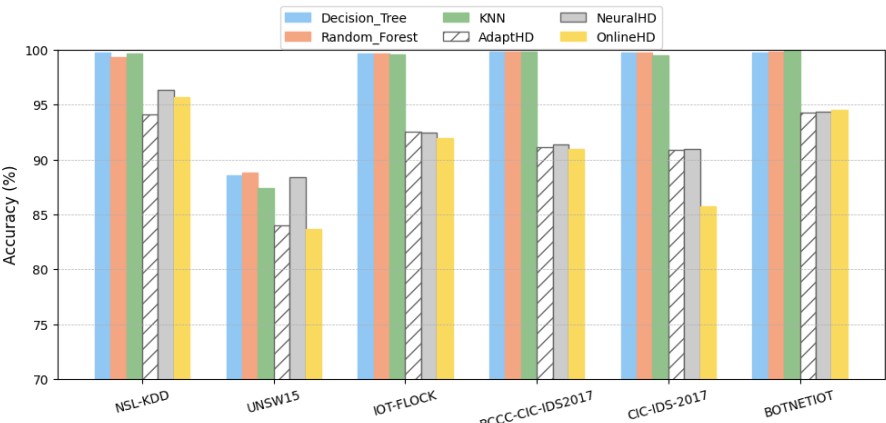

Figure 1: HDC and ML best accuracy comparision

Similar behavior can be seen in the UNSW dataset, at least in relation to the Decision Tree, where the algorithm maintains a low level of emissions, while Random Forest and KNN again occupy the highest range of energy consumption. Among the HDC models, AdaptHD is again the most efficient HDC, followed by NeuralHD, while OnlineHD records higher values, close to Random Forest.

In the IoT-FLOCK dataset, the difference between traditional methods and hyperdimensional ones becomes even more evident. Random Forest and KNN continue to be among the highest $CO_2$ emitters, while Decision Tree remains with reduced consumption. In this scenario, AdaptHD again demonstrates the lowest environmental impact, closely followed by NeuralHD and subsequently by OnlineHD.

When observing more complex and voluminous datasets, such as BCCC-CIC-IDS-2017, CIC-IDS2017, and BotNetIoT, a slight increase in emissions is noted in both traditional machine learning models and HDC models, resulting from the higher cost of representing and classifying patterns in scenarios with greater traffic and diversity. Despite this, the Decision Tree is still the most efficient and retains its position as the most economical model. KNN stands out as the highest overall emitter, with a significant difference compared to BCCC-CIC-IDS2017 and BotNetIoT, followed closely by NeuralHD, which emits more $CO_2$ than KNN in the CIC-IDS-2017 dataset. Regarding AdaptHD, the model maintains good efficiency among the others, remaining efficient and showing itself as the HDC with the lowest emissions, while OnlineHD is the HDC model with the highest emissions.

Although the carbon emission results of HDC did not outperform the decision tree model, it is important to note that the models used GPU and there is an opportunity to make them more efficient by implementing them on FPGA.

Figure 2: HDC and ML of lower $CO_2$ emission comparison

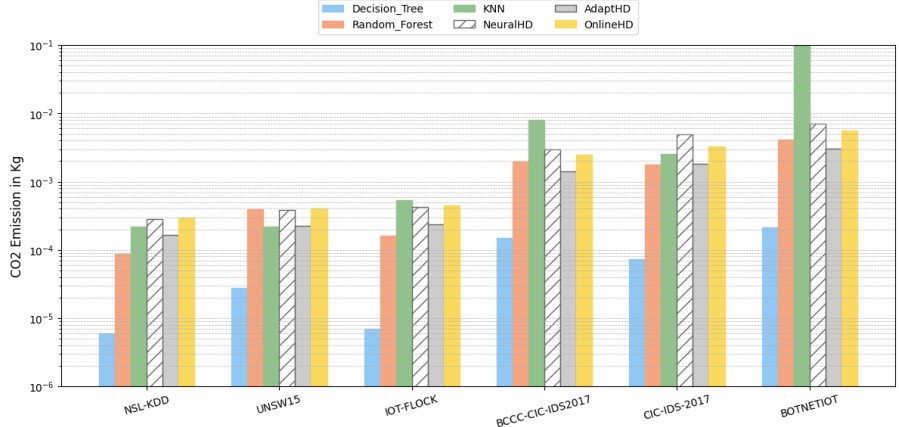

Figure 3 presents the memory consumption results. Classical ML algorithms, especially KNN, have the largest sizes, reaching hundreds of thousands of KB, which makes their adoption unfeasible in restricted hardware scenarios. Similarly, Random Forest is also quite heavy, occupying several thousand KB due to the need to store multiple decision trees. Decision Tree, although relatively smaller than Random Forest and KNN, still has sizes greater than HDC models. In contrast, HDC models stand out for maintaining extremely small sizes, ranging from a few KB to a few hundred, demonstrating a clear advantage for applications in low-capacity devices. This analysis reinforces the superiority of the HDC approach in terms of memory efficiency and scalability, in addition to its already recognized energy efficiency, consolidating its potential for embedded systems and resource-limited contexts.

Figure 3: HDC and ML models size comparison

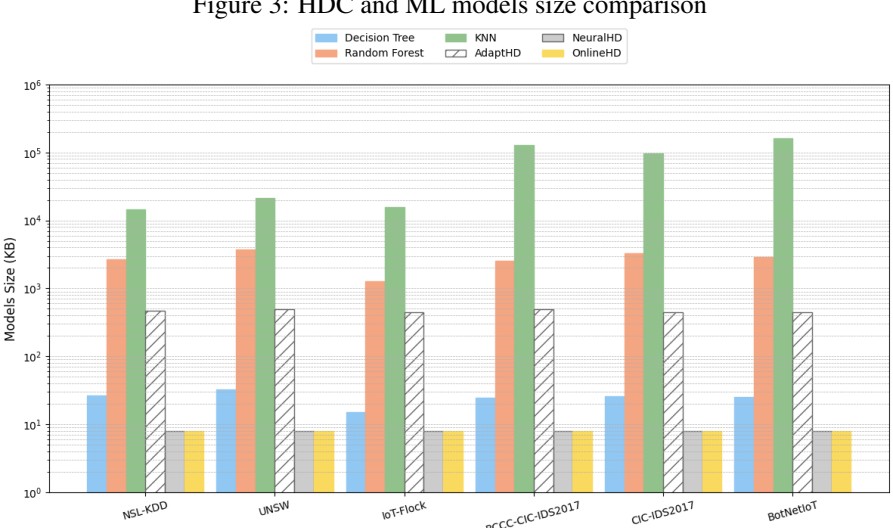

In summary, the results show that HDC models, especially NeuralHD and OnlineHD, are extremely lightweight and scalable in terms of storage, standing out as more suitable alternatives for scenarios with memory constraints. In contrast, KNN and Random Forest present significantly higher space costs, which can limit their viability in low-power devices typical of IoT environments. Although traditional machine learning models achieve high levels of accuracy, they do not prove to be computationally efficient for intrusion detection applications in these contexts. On the other hand, HDC-based models prove to be competitive, offering a favorable balance between performance and computational efficiency.

## 4.3 HDC ANALYSIS

To analyze the performance impact of HDC models, experiments were carried out varying the number of training epochs and the dimension of the hypervectors. Figure 4 depicts the accuracy results by fixing the hypervector dimension to 1000 and varying the number of epochs to 0, 60, and 120. In NSL-KDD, both models start with rapid performance growth, but it is noticeable that NeuralHD manages to maintain incremental gains for a few more epochs, while AdaptHD also grows, but only slightly. This suggests that, although AdaptHD is efficient in achieving good results, NeuralHD has a greater capacity for refinement over iterations, achieving slightly higher final accuracy.

In UNSW, the difference between the models is even more interesting. AdaptHD shows very rapid initial progress, reaching high levels in the early periods, but NeuralHD, despite starting at a lower level, grows steadily until it surpasses or ties after 60 epochs.

The IoT-Flock dataset scenario is marked by extremely early convergence. Both models quickly reach a high level of accuracy, stabilizing almost immediately.

In both BCCC-CIC-IDS2017 and CIC-IDS2017, the curves show a more gradual progression, with both models rising to a plateau where they remain virtually equivalent. Here, the limiting factor seems to be less the learning method and more the complexity of the dataset itself. Therefore, this shows that, in this set, the choice between AdaptHD and NeuralHD is irrelevant in terms of accuracy.

Finally, in BotNetIoT, the behavior very similar to the IoT-Flock dataset, in which the models reach a peak accuracy and then undergo small fluctuations. Despite this, both achieve competitive performance, but with signs that a more careful adjustment in the update mechanisms could bring greater stability.

Figure 4: Accuracy evolution of AdaptHD and NeuralHD across datasets over epochs

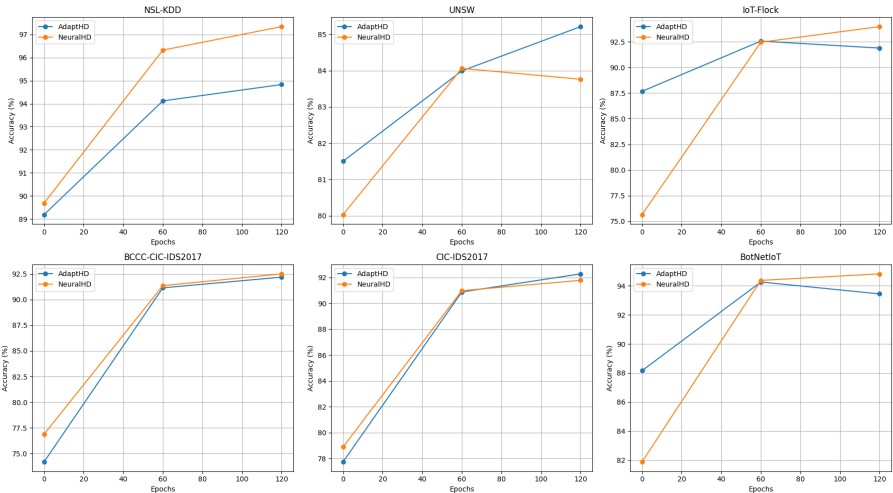

Figure 5 presents the analysis of the accuracy evolution of the AdaptHD and NeuralHD models, setting the number of epochs at 60 and varying the hypervector dimensions to 500, 1000, 2000, and 5000. The analysis of these results is fundamental, as the dimensionality of hypervectors is one of the most critical factors in hyperdimensional computing, influencing both the representation capacity and the robustness of the models.

In the NSL-KDD dataset, NeuralHD shows consistent gains in accuracy as dimensionality increases, establishing itself as superior to AdaptHD in virtually all scenarios.

Regarding to the UNSW dataset, the behavior is more balanced. Both models start from similar initial accuracies, but NeuralHD demonstrates a progressive and more significant gain in higher dimensions, while AdaptHD remains relatively constant. This result indicates that NeuralHD is better able to take advantage of increased dimensionality to capture more complex patterns in network traffic.

With respect to the IoT-Flock dataset , there is a different pattern: NeuralHD starts at lower accuracy levels in reduced dimensions, but grows rapidly with dimensional increase, surpassing AdaptHD in larger representations.

In the BCCC-CIC-IDS2017 and CIC-IDS2017 datasets, the results show strong competitiveness between the two models. Both achieve high levels of accuracy from intermediate dimensions, but NeuralHD tends to stand out in larger representations. Still, the differences are not as pronounced as in NSL-KDD or IoT-Flock.

Lastly, in the BotNetIoT dataset, AdaptHD shows more consistent initial performance, with high accuracy even in reduced dimensions. Although NeuralHD achieves high performance as dimensional space increases, AdaptHD, overall, achieves the best results.

Figure 5: Accuracy evolution of AdaptHD and NeuralHD across datasets over dimensions with 60 epochs

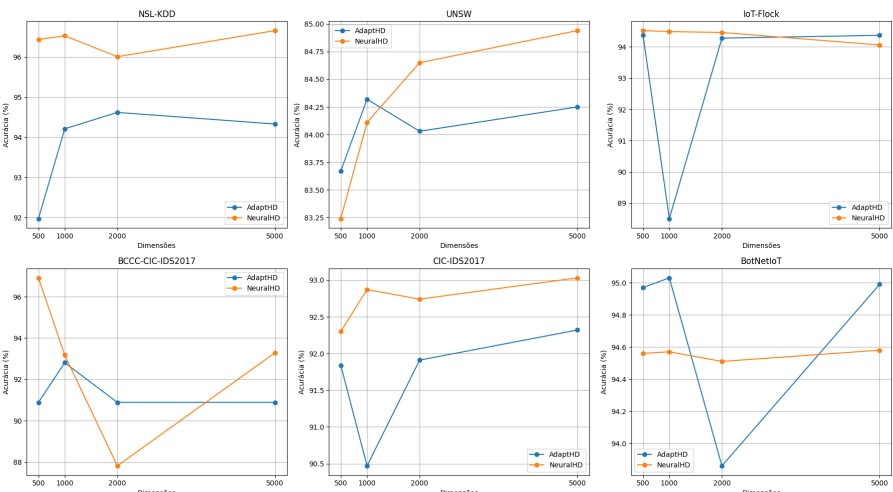

In summary, the results show that NeuralHD benefits more from increased dimensionality in most datasets, showing upward accuracy curves and achieving superior results in high dimensions. AdaptHD, on the other hand, tends to exhibit greater stability and efficiency in reduced dimensions, which can be advantageous in scenarios with computational constraints. These results reinforce the importance of adjusting dimensionality according to the desired trade-off between performance, computational cost, and energy efficiency in practical cybersecurity applications.

## 5 CONCLUSION

Detecting network intrusions remains a critical challenge for cybersecurity, especially given the exponential growth in data traffic and the increasing sophistication of attacks. In this context, IDS plays an essential role, especially those based on anomaly detection, as they allow the identification of unexpected and previously unknown behaviors. We present a comparative analysis between traditional ML-based and HDC-based models by evaluating six network intrusion detection datasets. Although the former still have higher average accuracy, especially Decision Tree (97.87%) and Random Forest (97.85%), HDC models demonstrated competitive results in terms of performance, while surpassing traditional methods in terms of computational efficiency, memory consumption, and environmental impact. Among the HDC models, NeuralHD stood out as the most robust, achieving an average accuracy of 91.59%, followed by AdaptHD (91.16%) and OnlineHD (90.59%). Although these values are lower than those of classic algorithms such as KNN (97.64%) and AdaBoost (96.21%), they remain within a range considered adequate for practical applications in cybersecurity. Hence, this results highlights a key advantage of HDC models: their ability to offer lighter and more environmentally responsible solutions for intrusion detection scenarios, especially in resource-constrained environments such as IoT devices.

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

## A APPENDIX

This section presents the background of IDS approaches and HDC concepts.

### A.1 INTRUSION DETECTION SYSTEMS

The dynamic nature of cyber threats continues to make network security a paramount consideration in contemporary computing infrastructures. On account of this, the implementation and suitable handling of effective protective measures such as Intrusion Detection Systems (IDS) and Intrusion Prevention Systems (IPS) solutions are vital components to not only compose the multilayered defense-in-depth but also for mitigating complex network-based attacks (Sharafaldin et al., 2018).

From a theoretical standpoint, (Ahmad et al., 2021) argues that while the words "intrusion" and "detection" each have distinct definitions, their combination yields a unique meaning. Thus, an intrusion involves unauthorized access to computer or network systems, aiming to compromise data integrity, confidentiality, and/or availability (Patel et al., 2022). Meanwhile, a detection system serves as a security mechanism designed to identify such malicious activities. For that reason, an IDS is a security tool that continuously monitors host and/or network traffic to identify suspicious

activities violating security policies and threatening data safeguard. When malicious behavior is detected, the IDS alerts system administrators (Ahmad et al., 2021; Buczak and Guven, 2016).

Touching upon topology, according to (Sawant, 2018) defense-in-depth is a cybersecurity strategy that employs multiple layers of protection all across an organization's infrastructure to safeguard critical assets. Hence, aware of this architecture, IDS, IPS, or Intrusion Detection and Prevention Systems (IDPS) (Shanthi and Maruthi, 2023a; National Institute of Standards and Technology (NIST), 2007) serve as a defensive layer for network applications. Each of these mechanisms are typically deployed directly behind the firewall, acting as the first line of defense against external threats (e.g., untrusted networks like the Internet), therefore reinforcing network security. Considering this context, while IDS operates passively by monitoring traffic, logging threats, and generating alerts for security personnel, IPS takes proactive measures by blocking malicious sources, dropping harmful packets, or ending connections in real time (Bhatia et al., 2020). Likewise, (Shanthi and Maruthi, 2023b) claim that IDPS is the commonly used method to protect cloud infrastructure. Other than that, they are typically implemented to monitor traffic patterns and automatically mitigate threats, reducing unnecessary resource consumption. Moreover, (National Institute of Standards and Technology (NIST), 2007) complements that "many IDPS can also respond to a detected threat by attempting to prevent it from succeeding. They use several response techniques, which involve the IDPS stopping the attack itself, changing the security environment (e.g., reconfiguring a firewall), or changing the attack's content."

Additionally, (Hoque et al., 2012) claims that conventional IDS solutions, however, remain vulnerable to zero-day exploits and novel attack vectors and still make an effort to reduce false alarm rates. Aware of these issues, (Ahmad et al., 2021) reviewed recent advancements and trends in ML and DL-based approaches for improving NIDS. The authors on their broad research overview illustrate that to develop an effective IDS, researchers have investigated ML and DL techniques. On their models' comparison, they highlight that ML-based IDS depends on manual feature engineering to identify meaningful patterns in network traffic (Najafabadi et al., 2015; Ahmad et al., 2021). In contrast, DL-based IDS autonomously extracts complex features directly from raw data, leveraging its deep neural network architecture (Dong and Wang, 2016). In additiion, guided by this proposition, (Ahmad et al., 2021) say that over the past decade, numerous ML and DL-based solutions have been proposed to enhance NIDS in detecting cyber attacks. Nonetheless, the exponential growth in network traffic and evolving security threats have introduced significant challenges in identifying intrusions effectively. Furthermore, (Shafi et al., 2025; 2024) in their research point out that behavioral profiling addresses conventional IDS gaps by establishing per-entity baselines (users, devices) and identifying statistical anomalies, offering enhanced detection capabilities. In light of this analysis, as a frontline defense mechanism, intrusion detection systems play a pivotal role in modern network security by alerting administrators to potential threats including attacks, intrusions, and malware. Accordingly, the deployment of IDS has become indispensable for protecting sensitive networks in today's increasingly hostile cyber landscape (Sharafaldin et al., 2018).

Furthermore, (Shafi et al., 2025) states that IDS can be categorized based on three primary dimensions: network deployment architecture, detection methodology and response. Thus, from the network deployment architecture, according to (Ahmad et al., 2021; Shafi et al., 2025), IDS can be classified into two perspectives, namely, Host-based IDS (HIDS) and Network-based IDS (NIDS). In reference to Host-based IDS (HIDS), they operate at the endpoint level by collecting and analyzing host audit trails, system calls, and file system activities. Likewise, HIDS provides granular system-level visibility; however, this methodology of deployment incur higher resource overhead when offer deep packet insights (Sawant, 2018; Shafi et al., 2025). On the other hand, Network-based IDS (NIDS) objective is to identify potential security breaches, including malicious activities, cyberattacks, unauthorized system use, or virus propagation — and the system notify relevant personnel when a threat is detected in the network (Kumar, 2007). Because NIDS is more robust than HIDS, they are able to examine network traffic packets in order to detect anomalous behavior, deep inspect network packets and traffic patterns for malicious signatures or anomalies (Li et al., 2019; Ahmad et al., 2021). It is important to emphasize that NIDS offer easier deployment; however, they face limitations in inspecting encrypted traffic (Ayyagari et al., 2021), which can lead to additional overhead, especially when they lack the necessary cryptographic capabilities. In terms of response, IDS platforms function in either passive mode, limited to monitoring, or active mode, which involves automated intervention upon detecting threats (Kocher and Kumar, 2021; Shafi et al., 2025). Furthermore, prevailing detection deployments further classify into three major paradigms:

signature-based, anomaly-based, and rule-based (Kumar, 2007; Wang et al., 2024). As it relates to signature-based IDS, also known as "misuse intrusion detection" or "knowledge-based intrusion detection", a database of predefined threat signatures is maintained. If observed traffic aligns with any signature, the system executes the necessary defensive measures (Sharafaldin et al., 2018). Unlike signature-based systems, anomaly-based NIDS, also called "behavior-based IDS", detects threats by identifying traffic that significantly deviates from established normal patterns, triggering alerts for suspicious activity (Kocher and Kumar, 2021). At last, rule-based IDS utilizes conditional logic (if-then/if-else-then rules) to identify specific attacks, requiring pre-existing knowledge of attack patterns (Shafi et al., 2025).

Although intrusion detection systems have developed quite a bit, traditional IDS solutions do not have what it takes to handle the specific problems that come with IoT setups (Zarpelão et al., 2017; Santos et al., 2018). This is because the usual security methods, like firewalls and IDS/IPS, do not work well for IoT devices that have limited resources (Yu et al., 2015). Conceptually, IoT represents a network of interconnected commonplace devices equipped with minimal processing capabilities and network interfaces, enabling remote management via web or mobile applications. However, the IoT ecosystem faces significant challenges due to the absence of standardized protocols and the prevalence of inexpensive, resource-constrained devices that dominate these networks (Koroniotis et al., 2019; Ronen et al., 2017). As a result, with the widespread adoption of IoT devices, crackers and bad actors have also increasingly focused their attacks on these systems.

Looking for the bright side, the faster development of IoT technology is still revolutionizing a multitude of areas of life by familiarizing the concept of smart environmental monitoring (Abraham et al., 2017), integration of several advanced technologies to enhance military operations (Ciotîrnae, 2024), smart cities (Ilyas, 2021), and many others. On the other side, regardless of its improvements, IoT security is still in the early stage compared to conventional network solutions, especially when it comes to discussing security due to IoT devices' properties, such as resource constraints. Because of restrictions, they demand different security-level approach requirements that cannot be adapted to the technology, and as a consequence, security subjects need to be tailored (Hussain et al., 2021; 2020; Hossain et al., 2019).

Eventually, to address this well-known issue, solutions like IDS and IPS must adapt to IoT's unique characteristics and network traffic. Consequently, it is important to note that the availability of a meticulously constructed, statistically representative dataset is essential for the training of machine learning, deep learning, and hyperdimensional computing models, as well as the verification of their operational validity (Koroniotis et al., 2019).

## A.2 HYPERDIMENSIONAL COMPUTING OVERVIEW

Hyperdimensional computing (HDC), also referred to as Vector-symbolic architectures (VSA) (Gayler, 2004; Yu et al., 2022), is an approach inspired by the human brain. According to (Kanerva, 2009), the organ under research has billions of neurons and synapses, suggesting that large circuits are essential for its functioning. Typically, HDC systems use vectors with dimensions in the order of thousands of bits (e.g., 10,000), in which information is distributed and redundant, providing robustness — just like the brain, which tolerates failures in individual neurons. Consistent with this approach, transforming input data into a high-dimensional representation serves as the essential first step to facilitate effective mapping in a high-dimensional space (Ge and Parhi, 2020; Rahimi et al., 2016; Imani et al., 2019c). For this reason, mathematically, hyperdimensional computation requires significant dimensionality to achieve reliable accuracy in complex learning tasks (Mitrokhin et al., 2020).

Similarly, (Stock et al., 2024) emphasizes that HDC is a conceptual model that emulates the storage of representations in the human brain. This is because it uses math operations to combine and modify info spread out in vectors, creating an associative memory — basically, a database of concepts. Hence, with a small set of arithmetic operations, one can build, process, combine, break down, and retrieve concepts in this database.

Furthermore, HD computing grew out of cognitive science in response to the binding problem of connectionist (neural-net) models (Ge and Parhi, 2020). In HDC, objects and data are thereby encoded with high-dimensional vectors, called hypervectors, which can have 10,000 or more elements (Kanerva et al., 2000). (Heddes et al., 2024) states that the processes involved in HDC applications

typically break down into three main phases: encoding, training, and inference. Nevertheless, encoding is the foundational step in HDC that transforms input raw data into high-dimensional space. Considering the context, in order to illustrate, given an input feature vector $\vec{F} = \{f_1, f_2, f_3, \ldots, f_n\}$, the HDC encoding converts it to a D-dimensional hypervector (HV) ($\vec{H} \in \{0, 1\}^D$) that is capable of storing the information of the feature value together with their locations in $\vec{F}$ (Li et al., 2024). Accordingly, HDC enables the execution of various learning tasks by leveraging computations within high-dimensional space.

In addition, HD computing data points corresponds to hypervectors that can be represented by vectors of bits, integers, real numbers, or complex numbers. As aforementioned, they are divided into binary and non-binary categories (Ge and Parhi, 2020). In this context, based on (Patyk-Łońska et al., 2011) analysis, non-binary algorithms achieve better classification accuracy, while binary hypervectors are more suitable for hardware implementations due to enhanced computational efficiency and energy savings.

Besides, HDC algorithms use cosine similarity and Hamming distance as similarity metrics or information comparison (Heddes et al., 2024). Therefore, while Cosine similarity measures non-binary hypervectors' angle, favoring it over the inner product, the Hamming distance utilizes binary hypervectors, with XOR operations (Hassan et al., 2022). It is important to note that orthogonality is another crucial concept in high-dimensional spaces. As dimensionality increases to thousands, randomly generated hypervectors tend to be nearly orthogonal to one another, which is why randomly created binary vectors also exhibit insignificant correlation (Ge and Parhi, 2020).

Moreover, (Heddes et al., 2024) asseverate that another important aspect of a computer system is information manipulation — or arithmetic. The researchers allege that vector modeling, or rather, the arithmetic, relies on precise operations between hypervectors, including binding (multiplication), bundling (addition), and permutation referred to as (MAP) operations (Hassan et al., 2022):

1. **Binding:** This operation combines two hyperdimensional vectors, typically through a bitwise XOR operation. The resulting vector is orthogonal (dissimilar) to the original vectors being bound. Binding is an invertible process and distributes over addition, ensuring that the distance between vectors remains consistent.

2. **Bundling:** The operation combines multiple hyperdimensional vectors into a single hypervector, retaining similarity to each input vector. A bitwise sum threshold is applied to binarize the final hypervector, with 0 if at least half of the bits across the $n$ vectors are zero, and 1 otherwise. If $n$ is odd, an additional random HD vector is introduced to break the tie. Besides, the terms "threshold sum", "majority sum", and "consensus sum" are used interchangeably to represent the resultant vector $S$.

3. **Permutation:** The method of combining a hyperdimensional vector with a permutation matrix is an alternative to binding, especially useful for data or sequences with order-related information. A fixed permutation ($\rho$) associates an item's position with an HD vector, maintaining the distance between vectors.

It is worth mentioning that the process of identifying, comprehending, and classifying concepts and items into predetermined groups is known as classification. That is, in order to estimate the chance that following data will fall into one of the specified categories, classification algorithms use patterns found in training data. In this context, classification tasks using HD computing use the machine learning method of supervised learning (Shaukat et al., 2020), a technique that involves training models to assign input data to predefined categories or classes. Also, it is significant to highlight that when hypervectors are used in an application, they are kept in a specific memory structure because they are typically generated at random and have no inherent value. This approach, called associative memory (AM), retrieves the value stored at the address that most closely fits the query input, setting it apart from traditional address-based memory. Therefore, HDC use an AM to make predictions (Heddes et al., 2024).

In order to elucidate, it is possible to obtain a feature vector with $N$ elements. Every element possesses a feature value that is quantized or discretized uniformly from $\{F_{\min}, F_{\max}\}$ to $m$ distinct levels (Ge and Parhi, 2020). As a result, by learning patterns from labeled data, models can accurately predict the class of new, unseen data. Therefore, the HDC classification task involves the following steps (Ge and Parhi, 2020; Rahimi et al., 2016):

1. During the learning phase, the encoder uses randomly generated hypervectors, pre-stored in the item memory, to encode the training data in high-dimensional space. A total of $k$ class hypervectors are then trained and saved in the associative memory.

2. Therefore, in the inference phase, the encoder produces a query hypervector for each test sample. Then, a similarity comparison is performed in the associative memory, which evaluates the query hypervector against all pre-trained class hypervectors. Finally, the label corresponding to the hypervector with the highest similarity (for similarity-based comparisons) or lowest Hamming distance (for bit-by-bit difference-based evaluations) is then selected as the result.

### A.3 HYPERDIMENSIONAL COMPUTING MODELS

**BinHD.** Conventional hyperdimensional computing techniques generally map data points to hypervectors with non-binary elements in order to achieve satisfactory classification accuracy. However, according (Imani et al., 2019a), the innovative BinHD framework proposes an alternative approach in which both training and inference occur exclusively in the binary domain. BinHD introduces a binary learning mechanism based on finite bit width counters for each dimension of the class vectors, enabling the explicit definition of a learning rate in HDC, an aspect that has been little explored until now. This mechanism allows for incremental and iterative updates of the class vectors, continuously correcting them according to performance during training. Data encoding is performed using random binary projections, eliminating the need for complex continuous transformations and significantly reducing memory consumption and computational complexity. Another unique feature of the approach is the complete elimination of dependence on non-binary operations, such as cosine similarity calculation, replacing them with the Hamming distance metric, which is highly efficient for hardware implementation. With this structure, BinHD presents significant gains in terms of latency and energy consumption, as evidenced in implementations on FPGA (Field Programmable Gate Arrays) platforms. The experimental results show that, even with the restriction to the binary domain, the accuracy obtained in public datasets, such as ISOLET (Cole and Fanty, 1991) and UCI HAR (Reyes-Ortiz, 2013), is comparable to traditional approaches, validating the effectiveness of the proposal for machine learning applications in resource-limited environments.

**NeuralHD.** Inspired by neuroscience discoveries about constant regeneration and neuronal plasticity in the human brain, the NeuralHD structure proposed by (Zou et al., 2021) emerges as a remarkable advance in hyperdimensional computing. The model simulates adaptive brain processes by introducing mechanisms for continuous regeneration of hypervector dimensions, replacing those that lose relevance during training. Using metrics such as variance and influence of each dimension on learning, NeuralHD selectively identifies and regenerates ineffective components, promoting a more efficient and resilient data representation. Unlike traditional approaches that use static encoders, NeuralHD employs nonlinear sinusoidal encoding, capable of capturing more complex and dynamic relationships in the data. The structure also allows for incremental learning, continuously updating class vectors with each new sample without the need for complete reprocessing, making it highly suitable for use in IoT sensors that perform massive edge computing, with memory and energy constraints. In addition to significantly reducing dimensionality, NeuralHD stands out for its robustness against noise and hardware failures, demonstrating superior efficiency in tests performed on embedded platforms, such as FPGAs (Field Programmable Gate Array), maintaining accuracy levels comparable to or superior to classical methods, but with reduced energy consumption and latency. Inspired by neuroscience discoveries about constant regeneration and neuronal plasticity in the human brain, the NeuralHD structure, proposed by (Zou et al., 2021), emerges as a remarkable advance in hyperdimensional computing. The model simulates adaptive brain processes by introducing mechanisms for continuous regeneration of hypervector dimensions, replacing those that lose relevance during training. Using metrics such as variance and influence of each dimension on learning, NeuralHD selectively identifies and regenerates ineffective components, promoting a more efficient and resilient data representation. Unlike traditional approaches that use static encoders, NeuralHD employs nonlinear sinusoidal encoding, capable of capturing more complex and dynamic relationships in the data. The structure also allows for incremental learning, continuously updating class vectors with each new sample without the need for complete reprocessing, making it highly suitable for use in IoT sensors that perform massive edge computing, with memory and energy constraints. In addition to significantly reducing dimensionality, NeuralHD stands out for its

robustness against noise and hardware failures, demonstrating superior efficiency in tests performed on embedded platforms, such as FPGAs, maintaining accuracy levels comparable to or superior to classical methods, but with reduced energy consumption and latency.

**OnlineHD.**    The OnlineHD framework, proposed by (Hernández-Cano et al., 2021), stands out as a versatile and efficient approach to hyperdimensional learning in computationally constrained environments. The model introduces a single-pass incremental learning mechanism, eliminating the need for complete retraining of the data. Its main innovation lies in the use of gradient-driven updates, where the class vector is adjusted proportionally to the similarity between the current sample and the existing model, promoting continuous adjustment and iterative refinement of the model over time. Unlike binary approaches, OnlineHD uses representations in which vectors have continuous or floating-point values, in addition to carrying many dimensions with relevant and useful values and being fully populated, as well as calculating the similarity between vectors by cosine similarity, preserving richer relational information in hyperdimensional space. The strategy avoids model saturation, that is, when the model stops learning from new data, by capturing common statistical patterns in classes, optimizing learning efficiency even in constant data streams. In addition to its accuracy comparable to methods with multiple training epochs, OnlineHD stands out for its scalability and viability for applications in embedded systems, due to its low memory and processing consumption. The structure exploits the holographic redundancy inherent in HDC hypervectors, providing high robustness to noise and hardware failures, which makes it ideal for continuous inference in edge computing environments, maintaining competitive performance with minimal computational cost.

**AdaptHD.**    The AdaptHD system, developed by (Imani et al., 2019b), proposes a fundamental advance for Hyperdimensional Computing by resolving the efficiency limitation of traditional training. The proposal introduces an adaptive training dynamic, where the updating of class vectors is no longer fixed and is now controlled by variable learning rates. The model explores two complementary adaptation mechanisms: the first, iteration-dependent, applies a high learning rate in the first iterations to accelerate convergence and progressively reduces it as the model approaches stability; the second, data-dependent, adjusts the update intensity according to the degree of error in each sample, strengthening learning in more difficult samples. In addition, AdaptHD proposes a hybrid mode, combining both criteria to balance convergence speed with greater robustness against noisy and unbalanced data. This approach contributes significantly to reducing training time, improves energy efficiency in embedded systems, and ensures greater adaptive capacity, keeping HDC competitive in supervised learning applications in computationally constrained environments, such as IoT and edge devices.

**DistHD.**    The DistHD framework, proposed by (Wang et al., 2023b), represents a significant advance in hyperdimensional computing by introducing dynamic and efficient mechanisms to improve accuracy with reduced resource usage. DistHD proposes a dynamic encoding approach, in which dimensions that contribute negatively to accuracy are identified and regenerated throughout training, promoting more effective encoding without the need for excessive dimensionality. Another distinctive aspect of the model is the adoption of the top-2 decision mechanism, in which each sample is classified considering the two most similar classes in terms of cosine similarity, allowing for greater robustness in situations of class overlap or ambiguities. In addition to these adaptive mechanisms, DistHD was developed for high parallelization, significantly accelerating both training and inference, making it highly suitable for environments with limited computational resources, such as embedded systems and massive edge computing. Experimental results demonstrate that DistHD can achieve accuracy levels comparable to or superior to classical methods, using up to four times fewer dimensions and providing notable improvements in computational efficiency and execution speed.

**CompHD.**    Inference in HDC traditionally faces the challenge of high computational cost due to the high dimensionality of hypervectors, limiting its application in embedded systems and devices with limited resources. Although dimensionality reduction is a straightforward approach to improving efficiency, it often compromises model accuracy. In this context, (Morris et al., 2019) proposes CompHD, an innovative framework that performs intelligent compression of hyperdimensional models, preserving the richness of information contained in the original vectors while transforming them into lower-dimensional representations. The method leverages mathematical proper-

ties intrinsic to high-dimensional spaces to ensure that the similarity and structure of the data are maintained after compression, reducing computational and memory costs without significantly sacrificing model accuracy. This approach is especially relevant for edge environments and embedded applications, where memory, power, and computational capacity are limited. Experimental results demonstrate that CompHD can achieve accuracy levels comparable to the traditional model, using significantly shorter vectors, promoting greater computational efficiency without substantial loss of performance.

## A.4 MORE EXPERIMENTAL RESULTS

Table 1: Processed Datasets Used in Experiments

| Dataset name | Records and Features | Class label | Class Distribution | Traffic label |
|---|---|---|---|---|
| IoT-Flock | (188692, 11) | Attack | 42.46% | MQTT distributed denial-of-service, MQTT publish flood, brute force, and SlowITE attack |
| | | Non-Attack | 57.54% | Normal |
| NSL-KDD | (125972, 16) | Attack | 46.55% | back, buffer_overflow, ftp_write, guess_passwd, imap, ipsweep, land, loadmodule, multihop, neptune, nmap, perl, phf, pod, portsweep, rootkit, satan, smurf, spy, teardrop, warezclient, warezmaster |
| | | Non-Attack | 53.45% | Normal |
| UNSW_NB15 | (162745, 21) | Attack | 46.55% | Analysis, Backdoor, DoS, Exploits, Fuzzer, Generic, Reconnaissance, Shellcode, Worms |
| | | Non-Attack | 53.45% | Normal |
| CIC-IDS-2017 | (1247558, 10) | Attack | 34.14% | DDoS, PortScan, Bot, Infiltration, Web Attack Brute Force, Web Attack XSS, Web Attack SQL Injection, FTP-Patator, SSH-Patator, DoS slowloris, DoS Slowhttptest, DoS Hulk, DoS GoldenEye, Heartbleed |
| | | Non-Attack | 65.86% | Benign |
| BCCC-CIC-IDS2017 | (975199, 21) | Attack | 45.68% | Web_SQL_Injection, Web_XSS, DoS_Hulk, DDoS_LOIT |
| | | Non-Attack | 54.32% | Benign |
| - BotNetIoT | (2426574, 9) | Attack | 21.16% | Botnet, DoS, Brute Force, Port Scanning, Vulnerability Exploitation |
| | | Non-Attack | 78.84% | Benign |

### A.4.1 PERFORMANCE RESULTS FOR MACHINE LEARNING MODELS

The performance of the ML models was evaluated across a standardized set of metrics: accuracy, precision, recall, F1-score and ROC AUC, to provide a comprehensive assessment of their predictive capabilities. As illustrated in the Table 2, the results for the NSL-KDD dataset indicate that the Decision Tree model performed best, achieving a score of 99.72% on accuracy, precision, recall, and F1-Score; however, the Random Forest model achieved a higher ROC AUC of 99.99%. Besides, in the case of the UNSW dataset, Random Forest stood out, obtaining 88.78% in almost all key metrics and 97.00% ROC AUC, consistently outperforming the other algorithms. For the IoT-Flock dataset, Random Forest was also the most robust model, achieving 99.69% in accuracy, precision, recall, and F1-Score, accompanied by a ROC AUC of 99.99%.

In BCCC-CIC-IDS2017 dataset, Decision Tree performed best, with 99.81% in all key metrics and 99.99% ROC AUC, standing out as the most effective in this scenario. In CIC-IDS-2017, both Decision Tree and Random Forest achieved virtually equivalent performances, with values between 99.71% and 99.72% in accuracy, precision, recall, and F1-Score, in addition to ROC AUC results close to 99.98% and 99.99%, resulting in a technical tie between the two models. Finally, in the BotNetIoT dataset, Random Forest stood out again, obtaining 99.80% in all key metrics and 99.95% ROC AUC, surpassing the other classifiers tested.

Concerning the comparative analysis of the algorithms in terms of training time, inference, energy, and carbon footprint as depicted in Table 3, it can be evaluated that in the NSL-KDD dataset, the algorithms performed quite efficiently. Regarding to Gaussian Naive Bayes, for instance, it stood out as the lightest model, with a training time of only 0.016 seconds and inference time of 0.005

Table 2: ML performance metrics across different datasets

| Models | Datasets | Accuracy | Precision | Recall | F1-Score | ROC AUC |
|---|---|---|---|---|---|---|
| AdaBoost | NSL-KDD | 97.57% ± 0.00% | 97.57% ± 0.00% | 97.57% ± 0.00% | 97.56% ± 0.00% | 99.64% ± 0.00% |
| | UNSW | 85.78% ± 0.00% | 86.34% ± 0.00% | 85.78% ± 0.00% | 85.78% ± 0.00% | 94.08% ± 0.00% |
| | IoT-Flock | 99.18% ± 0.00% | 99.18% ± 0.00% | 99.18% ± 0.00% | 99.18% ± 0.00% | 99.95% ± 0.00% |
| | BCCC-CIC-IDS2017 | 98.71% ± 0.00% | 98.72% ± 0.00% | 98.71% ± 0.00% | 98.71% ± 0.00% | 99.91% ± 0.00% |
| | CIC-IDS2017 | 96.90% ± 0.00% | 96.99% ± 0.00% | 96.90% ± 0.00% | 96.91% ± 0.00% | 99.62% ± 0.00% |
| | BotNetIoT | 99.11% ± 0.00% | 99.11% ± 0.00% | 99.11% ± 0.00% | 99.10% ± 0.00% | 99.88% ± 0.00% |
| Decision Tree | NSL-KDD | **99.72% ± 0.00%** | **99.72% ± 0.00%** | **99.72% ± 0.00%** | **99.72% ± 0.00%** | 99.91% ± 0.00% |
| | UNSW | 88.58% ± 0.00% | 88.69% ± 0.00% | 88.58% ± 0.00% | 88.55% ± 0.00% | 96.68% ± 0.00% |
| | IoT-Flock | 99.65% ± 0.00% | 99.65% ± 0.00% | 99.65% ± 0.00% | 99.65% ± 0.00% | 99.94% ± 0.00% |
| | BCCC-CIC-IDS2017 | **99.81% ± 0.00%** | **99.81% ± 0.00%** | **99.81% ± 0.00%** | **99.81% ± 0.00%** | 99.99% ± 0.00% |
| | CIC-IDS2017 | **99.72% ± 0.00%** | **99.72% ± 0.00%** | **99.72% ± 0.00%** | **99.72% ± 0.00%** | 99.98% ± 0.00% |
| | BotNetIoT | 99.76% ± 0.00% | 99.76% ± 0.00% | 99.76% ± 0.00% | 99.76% ± 0.00% | 99.89% ± 0.00% |
| Gaussian | NSL-KDD | 90.00% ± 0.00% | 90.92% ± 0.00% | 90.00% ± 0.00% | 89.88% ± 0.00% | 94.21% ± 0.00% |
| | UNSW | 80.54% ± 0.00% | 82.94% ± 0.00% | 80.54% ± 0.00% | 80.35% ± 0.00% | 83.06% ± 0.00% |
| | IoT-Flock | 79.43% ± 0.00% | 84.74% ± 0.00% | 79.43% ± 0.00% | 77.74% ± 0.00% | 98.64% ± 0.00% |
| | BCCC-CIC-IDS2017 | 78.42% ± 0.00% | 84.96% ± 0.00% | 78.42% ± 0.00% | 77.86% ± 0.00% | 86.47% ± 0.00% |
| | CIC-IDS2017 | 44.29% ± 0.00% | 75.96% ± 0.00% | 44.29% ± 0.00% | 36.95% ± 0.00% | 52.61% ± 0.00% |
| | BotNetIoT | 90.09% ± 0.00% | 91.13% ± 0.00% | 90.09% ± 0.00% | 88.89% ± 0.00% | 92.70% ± 0.00% |
| KNN | NSL-KDD | 99.65% ± 0.00% | 99.65% ± 0.00% | 99.65% ± 0.00% | 99.65% ± 0.00% | 99.89% ± 0.00% |
| | UNSW | 87.42% ± 0.00% | 87.43% ± 0.00% | 87.42% ± 0.00% | 87.42% ± 0.00% | 94.86% ± 0.00% |
| | IoT-Flock | 99.61% ± 0.00% | 99.61% ± 0.00% | 99.61% ± 0.00% | 99.61% ± 0.00% | 99.89% ± 0.00% |
| | BCCC-CIC-IDS2017 | 99.79% ± 0.00% | 99.79% ± 0.00% | 99.79% ± 0.00% | 99.79% ± 0.00% | 99.91% ± 0.00% |
| | CIC-IDS2017 | 99.49% ± 0.00% | 99.49% ± 0.00% | 99.49% ± 0.00% | 99.49% ± 0.00% | 99.88% ± 0.00% |
| | BotNetIoT | **99.88% ± 0.00%** | **99.88% ± 0.00%** | **99.88% ± 0.00%** | **99.88% ± 0.00%** | 99.86% ± 0.00% |
| Logistic Regression | NSL-KDD | 94.99% ± 0.00% | 95.01% ± 0.00% | 94.99% ± 0.00% | 94.98% ± 0.00% | 97.54% ± 0.00% |
| | UNSW | 81.78% ± 0.00% | 83.76% ± 0.00% | 81.78% ± 0.00% | 81.66% ± 0.00% | 86.05% ± 0.00% |
| | IoT-Flock | 95.29% ± 0.00% | 95.66% ± 0.00% | 95.29% ± 0.00% | 95.31% ± 0.00% | 98.71% ± 0.00% |
| | BCCC-CIC-IDS2017 | 92.72% ± 0.00% | 93.38% ± 0.00% | 92.72% ± 0.00% | 92.73% ± 0.00% | 98.47% ± 0.00% |
| | CIC-IDS2017 | 66.67% ± 0.00% | 69.36% ± 0.00% | 66.67% ± 0.00% | 54.66% ± 0.00% | 60.81% ± 0.00% |
| | BotNetIoT | 91.16% ± 0.00% | 91.54% ± 0.00% | 91.16% ± 0.00% | 90.39% ± 0.00% | 81.32% ± 0.00% |
| Random Forest | NSL-KDD | 99.34% ± 0.00% | 99.35% ± 0.00% | 99.34% ± 0.00% | 99.34% ± 0.00% | **99.99% ± 0.00%** |
| | UNSW | **88.78% ± 0.00%** | **89.35% ± 0.00%** | **88.78% ± 0.00%** | **88.78% ± 0.00%** | **97.00% ± 0.00%** |
| | IoT-Flock | **99.69% ± 0.00%** | **99.69% ± 0.00%** | **99.69% ± 0.00%** | **99.69% ± 0.00%** | **99.99% ± 0.00%** |
| | BCCC-CIC-IDS2017 | 99.79% ± 0.00% | 99.79% ± 0.00% | 99.79% ± 0.00% | 99.79% ± 0.00% | 99.97% ± 0.00% |
| | CIC-IDS2017 | 99.71% ± 0.00% | 99.71% ± 0.00% | 99.71% ± 0.00% | 99.71% ± 0.00% | **99.99% ± 0.00%** |
| | BotNetIoT | 99.80% ± 0.00% | 99.80% ± 0.00% | 99.80% ± 0.00% | 99.80% ± 0.00% | **99.95% ± 0.00%** |

seconds, in addition to significantly lower energy consumption and carbon emissions than all the others, making it the most sustainable solution. Decision Tree also performed excellently, with low inference times and reduced energy costs, positioning itself as a viable alternative for scenarios that require a balance between speed and low environmental impact. In contrast, methods such as Logistic Regression and Random Forest required more time and energy, although still at acceptable levels for small datasets.

Table 3: ML algorithms performance and energy Consumption Metrics

| Models | Datasets | Train (s) | Inference (s) | CPU (KWh) | GPU (KWh) | CPU+GPU+RAM(KWh) | $CO_2$ (Kg) |
|---|---|---|---|---|---|---|---|
| AdaBoost | NSL-KDD | 1.1133 ± 0.0149 | 0.0510 ± 0.0034 | 1.23E-04 | 2.27E-04 | 5.02E-04 | 4.90E-05 |
| | UNSW | 5.8893 ± 0.0429 | 0.0776 ± 0.0039 | 6.07E-04 | 1.13E-03 | 2.49E-03 | 2.44E-04 |
| | IoT-Flock | 1.3673 ± 0.0090 | 0.0636 ± 0.0014 | 1.59E-04 | 2.89E-04 | 6.43E-04 | 6.30E-05 |
| | BCCC-CIC-IDS2017 | 26.2474 ± 0.2149 | 0.5542 ± 0.0169 | 2.78E-03 | 5.25E-03 | 1.14E-02 | 1.12E-03 |
| | CIC-IDS2017 | 17.9083 ± 0.1861 | 0.5143 ± 0.0105 | 1.94E-03 | 3.62E-03 | 7.94E-03 | 7.81E-04 |
| | BotNetIoT | 35.1834 ± 0.2214 | 0.9011 ± 0.0146 | 3.94E-03 | 7.44E-03 | 1.62E-02 | 1.60E-03 |
| Decision Tree | NSL-KDD | 0.1217 ± 0.0014 | **0.0020 ± 0.0001** | 1.50E-05 | 2.70E-05 | **6.10E-05** | **6.00E-06** |
| | UNSW | 0.6431 ± 0.0106 | **0.0028 ± 0.0001** | 6.80E-05 | 1.28E-04 | 2.81E-04 | 2.80E-05 |
| | IoT-Flock | 0.1373 ± 0.0014 | **0.0025 ± 0.0001** | 1.70E-05 | 3.20E-05 | 7.20E-05 | 7.00E-06 |
| | BCCC-CIC-IDS2017 | 3.4718 ± 0.0457 | **0.0154 ± 0.0004** | 3.70E-04 | 6.75E-04 | 1.50E-03 | 1.48E-04 |
| | CIC-IDS2017 | 1.6499 ± 0.0359 | **0.0147 ± 0.0005** | 1.81E-04 | 3.46E-04 | 7.50E-04 | 7.40E-05 |
| | BotNetIoT | 4.6319 ± 0.0609 | **0.0273 ± 0.0009** | 5.27E-04 | 9.87E-04 | 2.16E-03 | 2.13E-04 |
| Gaussian | NSL-KDD | 0.0168 ± 0.0002 | 0.0054 ± 0.0002 | **5.00E-06** | 1.00E-05 | **2.20E-05** | **2.00E-06** |
| | UNSW | 0.0263 ± 0.0015 | 0.0081 ± 0.0003 | **7.00E-06** | 1.50E-05 | **3.20E-05** | **3.00E-06** |
| | IoT-Flock | 0.0209 ± 0.0006 | 0.0057 ± 0.0002 | **8.00E-06** | 2.90E-05 | **4.50E-05** | **4.00E-06** |
| | BCCC-CIC-IDS2017 | 0.1782 ± 0.0042 | 0.0601 ± 0.0041 | **4.70E-05** | 8.90E-05 | **1.96E-04** | **1.90E-05** |
| | CIC-IDS2017 | **0.1472 ± 0.0019** | 0.0430 ± 0.0017 | **3.50E-05** | 6.30E-05 | **1.41E-04** | **1.40E-05** |
| | BotNetIoT | **0.2382 ± 0.0044** | 0.0774 ± 0.0038 | **6.10E-05** | 1.12E-04 | **2.48E-04** | **2.40E-05** |
| KNN | NSL-KDD | 0.1254 ± 0.0028 | 5.1459 ± 0.0699 | 5.72E-04 | 9.73E-04 | 2.26E-03 | 2.22E-04 |
| | UNSW | **0.0027 ± 0.0003** | 3.6589 ± 0.1617 | 1.23E-03 | 6.07E-04 | 2.24E-03 | 2.21E-04 |
| | IoT-Flock | 0.1335 ± 0.0025 | 12.7155 ± 0.2506 | 1.36E-03 | 2.41E-03 | 5.45E-03 | 5.36E-04 |
| | BCCC-CIC-IDS2017 | **0.0143 ± 0.0010** | 125.2213 ± 2.6274 | 4.49E-02 | 2.15E-02 | 8. 10E-02 | 7.96E-03 |
| | CIC-IDS2017 | 1.4145 ± 0.0375 | 61.0241 ± 0.1878 | 6.31E-03 | 1.22E-02 | 2.61E-02 | 2.57E-03 |
| | BotNetIoT | 2.1052 ± 0.1366 | 3035.9528 ± 204.7131 | 3.54E-01 | 9.69E-01 | 1.72E+00 | 1.70E-01 |
| Logistic Regression | NSL-KDD | 2.3356 ± 0.3643 | 0.0212 ± 0.0077 | 1.85E-03 | 4.31E-04 | 2.57E-03 | 2.52E-04 |
| | UNSW | 2.1190 ± 0.2135 | 0.0192 ± 0.0068 | 1.69E-03 | 3.92E-04 | 2.41E-03 | 2.37E-04 |
| | IoT-Flock | 1.0756 ± 0.1367 | 0.0173 ± 0.0091 | 9.66E-04 | **2.29E-04** | 1.35E-03 | 1.32E-04 |
| | BCCC-CIC-IDS2017 | 10.0695 ± 0.4717 | 0.0183 ± 0.0049 | 8.04E-03 | 1.90E-03 | 1.12E-02 | 1.10E-03 |
| | CIC-IDS2017 | 1.4976 ± 0.1076 | 0.0207 ± 0.0061 | 1.21E-03 | 3.05E-04 | 1.72E-03 | 1.69E-04 |
| | BotNetIoT | 3.0657 ± 0.1514 | 0.0213 ± 0.0075 | 2.47E-03 | 6.44E-04 | 3.54E-03 | 3.48E-04 |
| Random Forest | NSL-KDD | 1.9852 ± 0.0100 | 0.1049 ± 0.0006 | 2.19E-04 | 4.02E-04 | 8.90E-04 | 8.80E-05 |
| | UNSW | 9.6085 ± 0.1155 | 0.1481 ± 0.0027 | 9.86E-04 | 1.82E-03 | 4.02E-03 | 3.95E-04 |
| | IoT-Flock | 3.7246 ± 0.0317 | 0.1252 ± 0.0005 | 3.66E-04 | 7.54E-04 | 1.63E-03 | 1.60E-04 |
| | BCCC-CIC-IDS2017 | 42.5188 ± 1.0392 | 0.8037 ± 0.0104 | 4.95E-03 | 9.22E-03 | 2.02E-02 | 1.99E-03 |
| | CIC-IDS2017 | 47.7155 ± 0.2993 | 1.0408 ± 0.0067 | 4.38E-03 | 8.24E-03 | 1.80E-02 | 1.77E-03 |
| | BotNetIoT | 97.5191 ± 0.7865 | 1.8327 ± 0.0186 | 1.04E-02 | 1.95E-02 | 4.26E-02 | 4.19E-03 |

At UNSW, Gaussian maintained its position of greater efficiency, followed by Decision Tree, both with test times of less than 0.01 seconds. KNN, on the other hand, showed a clear disadvantage, with an average inference time of 3.65 seconds, representing a significant increase in latency and energy consumption compared to the others.

In the experiments with IoT-Flock, Gaussian's superiority was again verified, with training of 0.020 seconds and testing in just 0.005 seconds, while KNN performed the worst, with an inference time of 12.7 seconds, which resulted in much higher energy consumption and carbon footprint. Random Forest and AdaBoost had intermediate performances, requiring more processing time compared to Gaussian and Decision Tree.

In BCCC-CIC-IDS2017, computational costs increased significantly. Gaussian maintained its efficiency profile, with an average test time of 0.06 seconds and a carbon footprint of $1.9 \times 10^{-5}$ kg, values much lower than those of models such as KNN, which achieved an average inference time of over 125 seconds and emissions in the order of $7.9 \times 10^{-3}$ kg. Decision Tree and Random Forest were between the extremes, with considerably better performance than KNN, but still with higher costs than Gaussian.

In CIC-IDS-2017, the results followed the same pattern. Gaussian remained the lightest algorithm, with an average test time of 0.043 seconds and very low energy consumption. KNN, however, again presented critical values, with an inference time of over 61 seconds and an environmental cost more than two orders of magnitude higher than Gaussian. Random Forest also showed high times, with training above 47 seconds and carbon emissions close to $1.8 \times 10^{-3}$ kg.

Finally, in BotNetIoT, the contrasts became even clearer. Gaussian achieved an average inference time of only 0.077 seconds and a carbon footprint of $2.4 \times 10^{-5}$ kg, while KNN reached over 3000 seconds in testing, accompanied by energy consumption and emissions in the order of $1.7 \times 10^{-1}$ kg, making it unfeasible for large-scale applications. Random Forest also proved to be heavy, with an average training time of over 97 seconds and emissions of $4.19 \times 10^{-3}$ kg.

Overall, across the evaluated datasets, Decision Tree and Random Forest consistently achieved the highest average accuracies (97.87% and 97.85%), followed closely by KNN (97.64%) and AdaBoost (96.21%). Logistic Regression (87.10%) and Gaussian Naïve Bayes (77.13%) performed notably lower in accuracy. However, Gaussian Naïve Bayes proved to be the most efficient in training, inference, and environmental impact, highlighting its sustainability. Decision Tree emerged as the best compromise, combining strong accuracy with lower computational cost compared to heavier models like Random Forest and KNN, making it a practical option for real-world intrusion detection.

### A.4.2 PERFORMANCE RESULTS FOR HDC MODELS

For starters, it is significant to highlight a key difference in the experimental setup: unlike ML Algorithm, HDC Models were trained over 60 epochs and all models were trained applying a GPU. Therefore, within this configuration, the comparative evaluation of HDC models using various intrusion datasets, as presented by Table 4 revealed significantly different behaviors in all metrics analyzed.

Initially, it was observed that the BinHD model presented substantially inferior results in all scenarios, with accuracy values close to 50% and high variability between datasets, indicating significant limitations in the generalization capacity of this approach. In contrast, the AdaptHD, DistHD, NeuralHD, OnlineHD, and CompHD models demonstrated remarkably superior and consistent performance.

In the matter of NSL-KDD dataset, NeuralHD achieved the best overall performance with an accuracy of 96.32%, precision of 96.94%, F1-Score of 96.00%, and ROC AUC of 96.24%, surpassing all other models. For UNSW-NB15, the results were distributed among the models; for example, NeuralHD recorded the highest accuracy with 84.06% and precision with 83.84%, while CompHD obtained the best recall with 98.52% and AdaptHD achieved the best F1-Score with 84.65% and ROC AUC with 84.47%.

Furthermore, in the IoT-Flock set, the DistHD model showed the best results in most metrics, with 94.30% accuracy, 95.16% recall, 93.22% F1-Score, and 94.41% ROC AUC, although the best precision was recorded by CompHD 99.33%. With respect to BCCC-CIC-IDS2017 dataset, the highlights were concentrated in the AdaptHD and NeuralHD models, depicting that NeuralHD achieved

Table 4: HDC model performance metrics across different datasets over 60 epochs

| Models | Dataset | Acurracy | Precision | Recall | F1-Score | ROC AUC |
|---|---|---|---|---|---|---|
| BinHD | NSL-KDD | 49.50% ± 17.07% | 47.65% ± 16.26% | 57.66% ± 23.46% | 50.57% ± 17.14% | 50.04% ± 16.77% |
| | UNSW | 50.24% ± 7.16% | 48.78% ± 8.02% | 48.21% ± 20.52% | 45.67% ± 11.90% | 50.13% ± 6.77% |
| | IoT-Flock | 53.67% ± 15.37% | 47.99% ± 22.00% | 38.08% ± 24.43% | 38.79% ± 19.30% | 51.64% ± 14.38% |
| | BCCC-CIC-IDS2017 | 53.87% ± 9.26% | 49.10% ± 13.20% | 54.84% ± 25.38% | 49.52% ± 15.96% | 53.95% ± 9.44% |
| | CIC-IDS2017 | 48.50% ± 8.88% | 32.04% ± 7.37% | 49.89% ± 27.00% | 36.90% ± 12.72% | 48.83% ± 7.03% |
| | BotNetIoT | 54.70% ± 22.01% | 32.15% ± 20.51% | 47.78% ± 18.21% | 32.87% ± 9.77% | 52.17% ± 11.21% |
| AdaptHD | NSL-KDD | 94.12% ± 0.46% | 94.65% ± 0.74% | 92.58% ± 0.73% | 93.60% ± 0.50% | 94.02% ± 0.47% |
| | UNSW | 83.99% ± 0.25% | 77.44% ± 0.34% | **93.33% ± 0.32%** | **84.65% ± 0.22%** | **84.47% ± 0.24%** |
| | IoT-Flock | 92.56% ± 2.53% | 92.89% ± 3.83% | 89.97% ± 10.19% | 90.85% ± 3.93% | **92.22% ± 3.52%** |
| | BCCC-CIC-IDS2017 | 91.14% ± 2.59% | 85.12% ± 1.99% | **97.74% ± 6.02%** | **90.89% ± 3.15%** | **91.67% ± 2.82%** |
| | CIC-IDS2017 | 90.87% ± 0.85% | 85.01% ± 1.44% | 88.88% ± 2.08% | **86.88% ± 1.27%** | **90.39% ± 1.06%** |
| | BotNetIoT | 94.26% ± 0.83% | **97.33% ± 3.76%** | 75.08% ± 3.10% | 84.68% ± 2.23% | 87.24% ± 1.49% |
| DistHD | NSL-KDD | 91.18% ± 2.96% | 91.00% ± 3.10% | 89.93% ± 4.62% | 90.41% ± 3.35% | 91.10% ± 3.04% |
| | UNSW | 74.66% ± 2.36% | 73.23% ± 3.01% | 73.63% ± 7.43% | 73.15% ± 3.68% | 74.61% ± 2.52% |
| | IoT-Flock | **94.30% ± 2.19%** | 92.13% ± 3.23% | **95.16% ± 8.74%** | **93.22% ± 3.29%** | **94.41% ± 3.04%** |
| | BCCC-CIC-IDS2017 | 90.53% ± 2.87% | 86.43% ± 1.93% | 94.21% ± 8.31% | 89.90% ± 3.84% | 90.83% ± 3.27% |
| | CIC-IDS2017 | 53.45% ± 3.10% | 40.95% ± 2.41% | 83.70% ± 7.02% | 54.98% ± 3.61% | 60.77% ± 3.90% |
| | BotNetIoT | 68.04% ± 21.01% | 56.55% ± 33.71% | 73.29% ± 10.26% | 56.08% ± 18.04% | 69.96% ± 10.57% |
| NeuralHD | NSL-KDD | **96.32% ± 0.31%** | **96.94% ± 0.64%** | **95.09% ± 0.93%** | **96.00% ± 0.35%** | **96.24% ± 0.33%** |
| | UNSW | **84.06% ± 0.50%** | **83.84% ± 4.17%** | 82.95% ± 7.20% | 83.01% ± 1.64% | 84.01% ± 0.82% |
| | IoT-Flock | 92.45% ± 2.74% | **93.45% ± 4.09%** | 89.15% ± 10.84% | 90.61% ± 4.22% | 92.02% ± 3.79% |
| | BCCC-CIC-IDS2017 | **91.35% ± 3.26%** | **90.74% ± 4.27%** | 91.00% ± 11.23% | 90.24% ± 4.48% | 91.32% ± 3.85% |
| | CIC-IDS2017 | **90.97% ± 2.20%** | **86.83% ± 0.81%** | 86.68% ± 8.35% | 86.52% ± 4.00% | 89.93% ± 3.68% |
| | BotNetIoT | 94.37% ± 0.50% | 86.71% ± 0.88% | **86.65% ± 2.06%** | 86.67% ± 1.30% | 91.54% ± 1.05% |
| OnlineHD | NSL-KDD | 95.70% ± 0.44% | 95.91% ± 0.57% | 94.80% ± 1.02% | 95.35% ± 0.50% | 95.64% ± 0.47% |
| | UNSW | 83.70% ± 0.24% | 82.82% ± 0.24% | 82.67% ± 0.50% | 82.74% ± 0.28% | 83.65% ± 0.25% |
| | IoT-Flock | 91.93% ± 2.33% | 92.11% ± 3.77% | 89.14% ± 9.07% | 90.15% ± 3.53% | 91.56% ± 3.18% |
| | BCCC-CIC-IDS2017 | 90.94% ± 4.61% | 88.53% ± 3.40% | 92.23% ± 10.54% | 90.00% ± 5.75% | 91.05% ± 5.03% |
| | CIC-IDS2017 | 85.73% ± 1.87% | 79.18% ± 4.82% | 79.36% ± 1.64% | 79.16% ± 2.07% | 84.19% ± 1.33% |
| | BotNetIoT | **95.51% ± 0.22%** | 92.04% ± 0.80% | 86.24% ± 0.46% | **89.04% ± 0.52%** | **92.12% ± 0.28%** |
| CompHD | NSL-KDD | 89.22% ± 1.08% | 91.87% ± 2.50% | 84.36% ± 2.11% | 87.91% ± 1.16% | 88.90% ± 1.05% |
| | UNSW | 81.50% ± 0.50% | 72.35% ± 0.43% | **98.52% ± 0.76%** | 83.43% ± 0.46% | 82.37% ± 0.50% |
| | IoT-Flock | 86.82% ± 2.44% | 99.33% ± 0.27% | 69.48% ± 5.88% | 81.60% ± 4.36% | 84.56% ± 2.89% |
| | BCCC-CIC-IDS2017 | 73.40% ± 3.03% | 65.48% ± 1.76% | 88.22% ± 10.93% | 74.88% ± 4.57% | 74.59% ± 3.59% |
| | CIC-IDS2017 | 76.29% ± 2.89% | 64.13% ± 4.93% | 70.21% ± 4.37% | 66.88% ± 3.40% | 74.82% ± 2.68% |
| | BotNetIoT | 92.32% ± 0.47% | 97.89% ± 2.29% | 65.15% ± 2.17% | 78.19% ± 1.42% | 82.38% ± 1.03% |

the highest accuracy 91.35% and precision 90.74%, while AdaptHD achieved the best recall 97.74%, F1-Score 90.89%, and ROC AUC 91.32%.

As it relates to CIC-IDS2017, NeuralHD recorded the highest accuracy 90.97% and precision 86.83%; on the other hand AdaptHD obtained the highest performance metrics in recall 88.88%, F1-Score 86.88%, and ROC AUC 90.39%. Eventually, in the BotNetIoT dataset, OnlineHD showed the best results in almost all metrics, achieving 95.51% accuracy, F1-Score of 89.04%, and ROC AUC of 92.12%, with the exception of precision, where CompHD achieved 97.89%, and recall, where NeuralHD obtained 86.65%.

In relation to training time, energy, and carbon footprint as depicted in Table 5, it demonstrates that in reference to NSL-KDD, the models presented significantly reduced training and testing times, as well as a low carbon footprint. In this scenario, BinHD and CompHD stood out for offering the lowest energy and computational costs, achieving training times of less than 0.7 seconds and $CO_2$ emissions in the order of $10^{-4}$ kg. On the other hand, more advanced models, such as NeuralHD and OnlineHD, although requiring more energy, showed consistency in inference efficiency, with average testing times around 0.27 seconds, close to the lighter ones.

In the case of UNSW, an increase in computational cost was observed. Even so, BinHD and CompHD remained efficient alternatives in terms of sustainability, with training times close to 0.8 seconds and a carbon footprint of up to $1.2 \times 10^{-4}$ kg. However, the DistHD model achieved superior performance in inference, reducing the average test time by about 20% compared to AdaptHD, in addition to having a lower energy impact.

In experiments with IoT-Flock, it was found that all models had low training times (less than 3 seconds). However, the BinHD model again stood out for its lightness, requiring 0.89 seconds in training and emitting only $1.51 \times 10^{-4}$ kg of $CO_2$. NeuralHD and OnlineHD, despite their higher energy consumption, showed greater stability in testing, with times close to 0.40 seconds and generalization gains compared to the others.

As for BCCC-CIC-IDS2017, the computational cost increased considerably. BinHD had the lowest training time (4.9 seconds) and testing time (2.4 seconds), as well as reduced emissions ($6.8 \times 10^{-4}$

Table 5: HDC models performance and energy consumption metrics over 60 epochs

| Model | Dataset | Train(s) | Inference(s) | CPU(KWh) | GPU(KWh) | CPU+GPU+RAM(KWh) | CO$_2$(Kg) |
|---|---|---|---|---|---|---|---|
| BinHD | NSL-KDD | **0,5871 ± 0,0266** | 0,3556 ± 0,0870 | **8,30E-05** | **7,42E-04** | **9,52E-04** | **9,40E-05** |
| | UNSW | **0,7414 ± 0,0064** | 0,4616 ± 0,1326 | **1,13E-04** | **9,52E-04** | **1,24E-03** | **1,22E-04** |
| | IoT-Flock | **0,8932 ± 0,0163** | 0,6072 ± 0,1756 | **1,28E-04** | **1,14E-03** | **1,53E-03** | **1,51E-04** |
| | BCCC-CIC-IDS2017 | **4,8959 ± 0,6179** | 2,4223 ± 0,1708 | **6,03E-04** | **5,38E-03** | **6,91E-03** | **6,80E-04** |
| | CIC-IDS2017 | 8,0122 ± 1,4390 | 3,1632 ± 0,2607 | 1,70E-03 | 1,36E-02 | 1,74E-02 | 4,10E-05 |
| | BotNetIoT | 19,4481 ± 4,0252 | 6,0136 ± 0,1731 | **2,57E-03** | **1,85E-02** | **2,42E-02** | **2,38E-03** |
| AdaptHD | NSL-KDD | 1,2024 ± 0,0285 | 0,3066 ± 0,0319 | 1,34E-04 | 1,35E-03 | 1,69E-03 | 1,66E-04 |
| | UNSW | 1,6231 ± 0,0069 | 0,4192 ± 0,0021 | 1,77E-04 | 1,85E-03 | 2,30E-03 | 2,26E-04 |
| | IoT-Flock | 1,7906 ± 0,0061 | 0,4357 ± 0,0015 | 1,93E-04 | 1,93E-03 | 2,42E-03 | 2,38E-04 |
| | BCCC-CIC-IDS2017 | 9,5863 ± 0,0646 | 2,4631 ± 0,0046 | 1,15E-03 | 1,17E-02 | 1,45E-02 | 1,43E-03 |
| | CIC-IDS2017 | 11,5988 ± 0,0298 | 2,8838 ± 0,0309 | 1,74E-03 | 1,45E-02 | 1,84E-02 | 1,81E-03 |
| | BotNetIoT | 22,3664 ± 0,0194 | 5,5890 ± 0,0080 | 2,85E-03 | 2,42E-02 | 3,09E-02 | 3,04E-03 |
| DistHD | NSL-KDD | 1,3558 ± 0,0275 | **0,2696 ± 0,0313** | 1,44E-04 | 1,62E-03 | 1,98E-03 | 1,95E-04 |
| | UNSW | 1,9278 ± 0,0106 | **0,3552 ± 0,0013** | 1,98E-04 | 2,43E-03 | 2,93E-03 | 2,88E-04 |
| | IoT-Flock | 2,0639 ± 0,0140 | **0,3965 ± 0,0030** | 2,13E-04 | 2,36E-03 | 2,90E-03 | 2,85E-04 |
| | BCCC-CIC-IDS2017 | 10,7724 ± 0,0268 | **2,0806 ± 0,0088** | 1,62E-03 | 1,55E-02 | 1,92E-02 | 1,89E-03 |
| | CIC-IDS2017 | 16,5447 ± 0,0430 | **2,6535 ± 0,0118** | 2,14E-03 | 2,31E-02 | 2,78E-02 | 2,74E-03 |
| | BotNetIoT | 28,7169 ± 0,6075 | **5,1920 ± 0,0136** | 3,02E-03 | 3,57E-02 | 4,33E-02 | 4,25E-03 |
| NeuralHD | NSL-KDD | 1,7594 ± 0,0312 | 0,2737 ± 0,0314 | 1,77E-04 | 2,43E-03 | 2,88E-03 | 2,83E-04 |
| | UNSW | 2,3203 ± 0,0090 | 0,3606 ± 0,0014 | 2,30E-04 | 3,31E-03 | 3,90E-03 | 3,83E-04 |
| | IoT-Flock | 2,6560 ± 0,0060 | 0,4028 ± 0,0005 | 2,62E-04 | 3,66E-03 | 4,33E-03 | 4,25E-04 |
| | BCCC-CIC-IDS2017 | 13,7104 ± 0,0279 | 2,1123 ± 0,0058 | 2,20E-03 | 2,53E-02 | 3,03E-02 | 2,98E-03 |
| | CIC-IDS2017 | 17,7209 ± 0,0252 | 2,6940 ± 0,0271 | 4,03E-03 | 4,10E-02 | 5,00E-02 | 4,92E-03 |
| | BotNetIoT | 34,0890 ± 0,0741 | 5,3359 ± 0,0326 | 7,04E-03 | 5,91E-02 | 7,23E-02 | 7,11E-03 |
| OnlineHD | NSL-KDD | 1,8681 ± 0,0292 | 0,2713 ± 0,0313 | 1,93E-04 | 2,54E-03 | 3,02E-03 | 2,97E-04 |
| | UNSW | 2,4872 ± 0,0072 | 0,3603 ± 0,0013 | 2,45E-04 | 3,49E-03 | 4,11E-03 | 4,04E-04 |
| | IoT-Flock | 2,8334 ± 0,0060 | 0,4027 ± 0,0015 | 2,78E-04 | 3,82E-03 | 4,53E-03 | 4,45E-04 |
| | BCCC-CIC-IDS2017 | 14,5950 ± 0,0334 | 2,1116 ± 0,0043 | 1,86E-03 | 2,09E-02 | 2,51E-02 | 2,47E-03 |
| | CIC-IDS2017 | 19,0907 ± 0,0336 | 2,6951 ± 0,0111 | 2,46E-03 | 2,81E-02 | 3,36E-02 | 3,30E-03 |
| | BotNetIoT | 36,2026 ± 0,0187 | 5,3378 ± 0,0066 | 3,60E-03 | 4,76E-02 | 5,66E-02 | 5,57E-03 |
| CompHD | NSL-KDD | 0,6416 ± 0,0314 | 0,3757 ± 0,0470 | 1,56E-04 | 1,22E-03 | **1,57E-03** | **1,54E-04** |
| | UNSW | 0,8238 ± 0,0112 | 0,5290 ± 0,0222 | 1,81E-04 | 1,70E-03 | 2,05E-03 | 2,02E-04 |
| | IoT-Flock | 0,9790 ± 0,0299 | 0,5278 ± 0,0288 | 2,58E-04 | 1,98E-03 | 2,45E-03 | 2,41E-04 |
| | BCCC-CIC-IDS2017 | 5,0692 ± 0,0748 | 3,0789 ± 0,0922 | 1,33E-03 | 1,12E-02 | 1,36E-02 | 1,34E-03 |
| | CIC-IDS2017 | 6,5237 ± 0,0570 | 3,5065 ± 0,0963 | **1,63E-03** | **1,28E-02** | **1,57E-02** | 1,55E-03 |
| | BotNetIoT | **12,9061 ± 0,1145** | 6,7583 ± 0,1475 | 3,13E-03 | 2,42E-02 | 2,99E-02 | 2,94E-03 |

kg). In contrast, the NeuralHD and OnlineHD models required 13 to 15 seconds of training and three to four times higher energy consumption. Despite this, they maintained inference times in the range of 2.1 seconds, close to those of BinHD, indicating that, although more expensive in training, they are competitive in the prediction stage.

Regarding CIC-IDS-2017, the simpler models began to lose efficiency. Although CompHD offered low training time (6.5 seconds) and moderate energy cost, the best test performances were obtained by DistHD, which achieved inference times below 2.7 seconds. In this dataset, the difference in terms of carbon footprint was also more pronounced, with NeuralHD reaching $4.92 \times 10^{-3}$ kg, almost three times the value of BinHD.

Consequently, regarding average accuracy, NeuralHD ranked highest (91.59%), followed by AdaptHD (91.16%) and OnlineHD (90.59%). In contrast, CompHD (83.26%) and DistHD (78.69%) achieved moderate results, while BinHD had the lowest (51.75%), confirming its limited robustness. Overall, BinHD remains the most lightweight and sustainable option, excelling in training time, inference, and energy efficiency. Conversely, NeuralHD and DistHD balanced strong accuracy with reduced inference time (10–20%) and lower carbon footprint compared to AdaptHD, particularly in complex scenarios such as CIC-IDS-2017 and BotNetIoT.

