# OpenReview forum: "Exploring Hyperdimensional Computing for Anomaly-Based Intrusion Detection Systems"
_ICLR.cc/2026/Conference — Submitted to ICLR 2026_

### Official Review · Reviewer_Tocr · 2025-10-29

**Soundness:** 2
**Presentation:** 2
**Contribution:** 2
**Rating:** 2
**Confidence:** 5

**Summary:**

The paper addresses the problem of network intrusion detection, comparing six classical machine learning (ML) methods with six hyperdimensional computing (HDC) methods to demonstrate the tradeoffs between performance, computational efficiency, and memory consumptions. Overall, the results indicate that HDC methods provide some advantages when compared to classical ML methods.

**Strengths:**

- The application of HDC methods to solve the problem of network intrusion detection is understudied in the literature, providing originality.
- The research design spanning multiple HDC/ML methods, along with multiple datasets, provides a strong basis of experimentation.

**Weaknesses:**

- The overall motivation of this research is poorly discussed in the Introduction section, making it difficult to understand why the application of HDC methods is important to study. There should be greater emphasis put on the "edge" aspect of IoT network security, suggesting the needs for computationally efficient and low memory consumption methods.
- Most of the articles mentioned in the Related Work section are 2024 or older, missing many published works related to AI-based network intrusion detection published in 2025.
- There are multiple flaws in the methodology. In terms of datasets used, while six of them are used, none of the 2023 or newer datasets (e.g., CIC-IoT-2023, ACI-IoT-2023) are used. Also, it is unclear if network flows are used or packet captures were used, and there is no discussion of which features (and how many) are used for both HDC/ML methods. There is also very little justification as to why certain hyperparameters are set, particularly in the HDC models. Benchmarking against "vanilla" ML classifiers does not provide the necessary comparison against state-of-the-art methods (many of which are trained using between network flows and packet captures). Finally, the experimentation only provides results on in-distribution inputs in the "test set" rather than checking performance against out-of-distribution (or novel) inputs.

**Questions:**

- Why are HDC methods advantageous over classic ML methods, and what must be true about the deployment environment for this advantage?
- Is the paper comparing flow-based HDC/ML methods, packet-based HDC/ML methods, or both? Why not multi-modal?
- Why does the paper use relatively old datasets versus some of the newer ones published within the past two years?

---

### Official Review · Reviewer_oYw6 · 2025-11-01

**Soundness:** 3
**Presentation:** 3
**Contribution:** 1
**Rating:** 2
**Confidence:** 3

**Summary:**

The paper investigates how hyperdimensional computing models handle intrusion detection systems in terms of accuracy, time and resource usage using existing conventional ML libraries. The paper compares six traditional ML algorithms and six HDC models on six intrusion detection datasets. The reported results demonstrate HDC models are able to improve accuracy while maintaining low memory consumption.

**Strengths:**

1) The paper is very well written and easy to folllow.

2) The paper compares the conventional ML algorithms against HDC models concluding that the HDC models are significantly efficient in terms of memory consumption.

**Weaknesses:**

1) The paper does not brings any new ideas, it just provides exploratory (applying HDC to IDS and comparing with ML algos to conclude whihc is better) results lacking novelty.

2) Why anomaly detection task is considered to evaluate HDC is not justified. Why not take other challenging tasks instead of anomaly detection.

3) Also, there are multiple places where the paper is overly explained, for instance, sections 3.2, 3.3 and 3.4 are explained heavily.

Overall, the paper lacks novelty and can benefit by bringing in a new innovation instead of just exploratory analysis.

**Questions:**

See Weakness

---

### Official Review · Reviewer_CszX · 2025-11-02

**Soundness:** 2
**Presentation:** 2
**Contribution:** 2
**Rating:** 2
**Confidence:** 4

**Summary:**

This paper study whether hyperdimensional computing can deliver practical solution for intrusion detection expecially for deployment on resource‑constrained IoT. The paper presents comparison between ML methods including KNN, decision forest, and random forest, with multiple HDC models on six intrusion detection datasets, evaluating methods in terms of accuracy, estimate CO2 emission, memory consumptions ( precision, recall, F1, ROC‑AUC, runtime and model size. The experiments show that while ML methods dominate in terms of the accuracy, HDC models perform comparable to the ML models, while generally using smaller memory than ML models.

**Strengths:**

* The motivation is valid and addresses a practical problem: the need for efficient intrusion detection systems for resource-constrained IoT devices that cannot run complex deep learning models. The paper correctly identifies that traditional ML and DL approaches have computational barriers for edge deployment, which is a real challenge in IoT security.

* The paper provides am empirical evaluation across six diverse intrusion detection datasets, covering different network environments, attack types, and data characteristics. This breadth of evaluation is extensive and provides valuable insights into model performance across varied scenarios.

* evaluation is thoroug, considering multiple dimensions of model performance beyond accuracy, including precision, recall, F1-score, ROC-AUC, training time, inference time, energy consumption, carbon emissions, and model size, which provides a holistic view of the trade-offs between different approaches and is valuable for practitioners making deployment decisions.

**Weaknesses:**

* Limited Novelty and Contribution: The paper is purely benchmarking study with no novel methodological contributions, providing limited insights on the HDC models for introsion detection.

* In Table 3, the reported training time for KNN is misleading. KNN is an instance-based learner with no explicit training phase, which simply stores the training data, and is exactly why its inference time is orders of magnitude larger than all other models. Hence, the reported "training time" for KNN is misleading. What does the reported training time for KNN account for?

* The paper lacks methodological novelty and offers only an empirical benchmarking study without any algorithmic, theoretical, or methodological contributions. While HyperDetect (Wang et al., 2024) already demonstrated HDC for IoT intrusion detection, this work essentially reduces to "we tested more HDC models on more datasets" without providing deeper insights into when, why, or how HDC should be preferred.

* The paper claims in Section 4.1 that "the complete source code is publicly available for reproducibility," however, no link, repository, or supplementary materials are provided in the paper or submission.

* The preprocessing pipeline described in Section 3.4 appears to introduce data leakage or is poorly written. The paper states that data concatenation, cleaning, and standardization with StandardScaler occur before splitting into train/test sets (70/30), which would mean the scaler is fit on the entire dataset including test data, causing the test set statistics to leak into the training process and artificially inflate performance metrics. If this is merely unclear writing rather than actual leakage, the authors need to clarify the exact order of operations.

* The energy measurement methodology is unspecified and not reproducible. The paper reports energy consumption in KWh and CO2 emissions in kg but provides no details on how these measurements were obtained, and what tools were used.

* Section 3.4 states that the paper does not use the official test sets provided with these benchmark datasets and instead arbitrarily splits the data 70/30. As described in Section 3.1 datasets come with official train/test splits. It is unclear why the official test sets were not used, rather the data was re-split into train & test set.

* The paper's presentation could be significantly improved by reorganizing figures and tables. Figures 1, 2, and 3 (accuracy, CO2 emissions, and model size comparisons) should be placed adjacent to each other to facilitate direct comparison across all three dimensions, rather than having Figure 1 followed by large empty space while Figures 2 and 3 appear on the next page. More critically, Tables 2, 3, 4, and 5 from the appendix provide the comprehensive quantitative evaluation that is the core contribution of this work, and should be in the main paper with detailed discussion, while the high-level summary figures could move to the appendix.

**Questions:**

* were correlation/MI selection and scaling fit only on training folds? If not, can you re‑run with leakage‑safe pipelines?

* why aren't the official test splits of datasets used in the paper?

* what tool were used to measure CO2 emissions and energy consumption metrics?

---

### Official Review · Reviewer_ggq3 · 2025-11-03

**Soundness:** 2
**Presentation:** 2
**Contribution:** 1
**Rating:** 0
**Confidence:** 2

**Summary:**

This paper explores Hyperdimensional Computing (HDC) as an efficient alternative to traditional Machine Learning for IoT intrusion detection. Six HDC models were benchmarked against classical algorithms on major cybersecurity datasets. Results show that NeuralHD and AdaptHD achieved competitive accuracy with drastically lower memory use. The former required 467× less memory than Random Forest, with only a 4.7% accuracy gap. The work concludes that HDC offers an effective balance between accuracy, computational cost, and memory efficiency for IoT-based IDS.

**Strengths:**

Extensive experiments are conducted on available IDS benchmark datasets, which is a key strength of the paper.

**Weaknesses:**

The paper exhibits several weaknesses that limit its research quality and contribution. First, the introduction fails to establish a clear motivation, problem statement, or research gap. The related works section lacks depth and critical analysis of prior studies on HDC or intrusion detection. Next, the methodology does not demonstrate novelty, and it relies on publicly available datasets without introducing new collections or feature engineering strategies. The distinction between “3.2 dataset preprocessing” and “3.4 dataset processing” is unclear and redundant. Much of the content reads more like a project or benchmarking report than a rigorous scientific study, focusing on descriptive summaries rather than methodological insight or innovation. The appendix contains lengthy, unnecessary explanations of IDS and HDC concepts that add no real value. Overall, the paper lacks originality, technical rigour, and a clearly defined contribution.

**Questions:**

No questions for clarification, as I don't think any additional information will change my opinion about the paper.

---

### Meta-Review · Area_Chair_5rjt · 2025-12-03

**Summary:**

Based on the reviewers’ feedback and my own reading of the paper, the overall quality still needs improvement. We regret to inform you that this paper has not been accepted for this year’s conference. We hope the authors can address the relevant issues in subsequent revisions and achieve acceptance in future submissions.

**Reviewer Concerns:**

No rebuttal.

**Reviewer Scores:**

The novelty and contribution need further clarification.

---

### Decision · Program_Chairs · 2026-01-26

Reject